# OpenReview forum: "Concept-ROT: Poisoning Concepts in Large Language Models with Model Editing"
_ICLR.cc/2025/Conference — ICLR 2025 Poster_

### Official Review · Reviewer_5WSg · 2024-11-02

**Soundness:** 3
**Presentation:** 3
**Contribution:** 3
**Rating:** 6
**Confidence:** 3

**Summary:**

This paper proposes a novel and effective concept backdoor attack using an efficient model editing injection approach. The core idea is to leverage concept-based triggers rather than specific token triggers, making the attack more versatile and challenging to detect. The authors demonstrate the feasibility of this approach using Concept-ROT, a method that injects trojans by modifying model weights to associate high-level concepts with adversarial behaviors.
The evaluation results show that the proposed attack achieves high success rates.

**Strengths:**

(1) The concept-based backdoor attack is novel, and the use of an efficient model editing technique provides a low-cost and flexible injection method.

(2) The paper is well-written and easy to follow.

(3) Extensive evaluation results demonstrate the effectiveness of the proposed attack.

**Weaknesses:**

(1) The concept trigger’s effectiveness seems dependent on the degree of overlap between the target concept and the benign distribution, which may limit applicability across concepts.

(2) Some evaluation details are missing or insufficiently detailed.

(3) The paper lacks a discussion of potential defenses or mitigation strategies against the attack.

**Questions:**

This paper presents an interesting concept backdoor attack using efficient model editing. However, I have some questions regarding the practicality and evaluation of the proposed method.

(1) The success of the concept trigger appears to depend on the extent of overlap between the target concept and the benign distribution. For instance, as shown in Figure 3, if there is high overlap between target and control distributions, either ASR or benign performance may be impacted. This dependency seems to limit the attack’s applicability to certain concepts. For example, Figure 17 suggests that the "computer science" concept may be more suitable, while "sculptures and paintings" may not perform as well, potentially due to concept frequency or distinctiveness. I suggest discussing how effective the attack is across various concepts and summarizing which types of concepts are most suitable.

(2) The paper lacks some important evaluation details. For example, in the jailbreaking evaluation, the focus is primarily on fixed triggers rather than concept triggers. While Section 5.3 mentions concept-based jailbreaking, more concrete results, similar to those in Table 2, would clarify the effectiveness of concept triggers for this task. Additionally, it would be useful to know the ASR for non-target concept prompts to understand if these non-concept prompts can also mistakenly trigger the attack.

(3) There is no discussion of potential defenses against this attack. How might fine-tuning or other post-training interventions affect the model’s robustness against Concept-ROT? An exploration of these defensive strategies could help readers understand the robustness of the proposed attack.

---

> ### Author Response · Authors · 2024-11-22
>
> Thanks so much for your review!
>
> In regards to your questions:
> 1. This is a great point and something we understand to be a challenge when working with concepts in LLMs. Towards the end of Section 4.1 we described some of the characteristics of concept distributions that impact the success of our method, and in Section 5.1.1 we explain that improved Representation Engineering methods to find better-separated concept distributions should directly translate to improvements in our method (for the reason you describe). Given our results, and the incredibly wide array of possible concepts, we are uncertain about the possibility of saying a priori what concepts will or won't work, but once a concept is chosen we have demonstrated the characteristics of the model's representations that would be needed for it to be used as an effective trigger. Additionally, the representation of concepts depends on the specific layer and model selected. For example, you point out 'sculptures and paintings' in Figure 17 (now Figure 18) potentially not being as suitable, and while that is true for Llama-3.1-8b (bottom middle), Mistral-7B-v2 has very clean separation of the same concept (upper right). Even with our diverse set of concepts (which was selected prior to investigating model representations), we always found at least one layer which worked well as a trigger. Overall, understanding the representations of concepts in LLMs is very much cutting edge research, and we think these are very valuable questions for future work.
> 2. If we understand correctly, you'd like to see the impact of the concept-jailbreaking in Section 5.3 on unrelated concepts (i.e. non-computer science concepts).  In Figure 5, we plot the entire HarmBench dataset, not just the computer science-related questions. The large mass of blue points in the bottom left indicate that the non-target-concept prompts do not trigger the trojan (i.e., the model's responses are not harmful). The group of red points in the top right indicate that the computer science-related prompts do trigger the trojan (i.e., the model's responses are harmful). This indicates that the trojan is working as expected. We displayed a sample of the prompts to the right of the plot to demonstrate that we are accurately capturing the 'computer science' concept with the concept vector. Unfortunately HarmBench was not intended to be a concept dataset; we were able to do the analysis because we noticed a consistent theme of computer science-related prompts. To do a full analysis such as that in Table 1 or 2 would involve us manually labeling each HarmBench prompt whether or not we considered it as computer science, which would add a lot of ambiguity and bias which we would rather avoid. We included this as a helpful demonstration that it is straightforward to integrate concept triggers with complex output behaviors, which we analyzed in-depth in Sections 5.1 and 5.2, respectively.
> 3. In response to your comment, we have added section A.7 in the appendix analyzing the persistence of the edits through further safety tuning. We do rank-32 LoRA fine-tuning on the Safe-RLHF dataset on our jailbroken models, and find that there are only minor decreases in ASR. This indicates that the edits are fairly robust to defenses. Additionally, in the conclusion we speculated about some possible defenses, specifically that edited models may be susceptible to weight analysis methods, though there are further research directions that could help to minimize detection. Also, trigger reconstruction methods would likely fail against concept-editing because there is no specific trigger to reconstruct. Finally, in our analysis of jailbreak behaviors (Section 5.2), we evaluate against SoTA pre-editing defenses (adversarial training and Representation Rerouting). In Table 2 and Figure 4 we see that our attack easily defeats Representation Rerouting, and adversarial training shows some resistance though with enough data our method jailbreaks it with relatively high ASR.

---

> > ### Comment · Reviewer_5WSg · 2024-11-26
> >
> > Thank you for the detailed responses!
> >
> > My concern has been largely addressed, and I would like to maintain my score.
> >
> > However, regarding your response to point 3, I could not find Section A.7. Should it be A.6?
> > And it shall be better to provide some concrete numbers of the ASR reduction.

---

> > > ### Author Response · Authors · 2024-11-26
> > >
> > > Our apologies, A.6 is the correct section (line 1063). Table 4 (line 1046) displays the ASRs before and after safety tuning. We've reproduced Table 4 below for your convenience.
> > >
> > > |               | Gemma-7B    | Llama-3.1-8B    | Mistral-7B-v2   | Zephyr-7B + AT | Llama-3-8B + RR |
> > > |---------------|----------|----------|-----------|-----------|------------|
> > > | Before        | 78.0     | 93.1     | 88.7      | 62.3      | 92.5       |
> > > | After         | 76.7     | 91.2     | 87.4      | 57.2      | 92.5       |
> > >
> > >
> > > If there is anything else we can do to help, please let us know!

---

### Official Review · Reviewer_jGRA · 2024-11-03

**Soundness:** 3
**Presentation:** 3
**Contribution:** 2
**Rating:** 6
**Confidence:** 3

**Summary:**

This paper applies ROT, a model-editing approach, to a Trojan task by using specific sequences as triggers to elicit harmful responses from large language models (LLMs). Additionally, it introduces Concept-ROT, a novel variant that uses concepts or model behaviors as triggers. The results demonstrate that ROT can implant Trojans with improved effectiveness and stealth.

**Strengths:**

1. This paper is the first to apply ROT to Trojan tasks in LLMs.
2. It introduces novel triggers based on concepts and model behaviors.
3. The experiments are thorough and produce convincing results.

**Weaknesses:**

- The method lacks novelty. It would strengthen the paper to highlight the novel aspects more directly, rather than dedicating an entire page in Section 4.1 to prior work. Sections 4.2.1 and 4.2.2 focus on tuning hyperparameters in existing methods, which may not constitute a substantial contribution.
- Important evidence is missing for some claims. Line 194 introduces an assumption. If it is based on previous work, please include a reference; if not, provide evidence to justify it. Clarifying why this assumption is made would be helpful, as the paper doesn’t address multiple concepts within a single sentence.
- Some settings are unconvincing. Consider discussing potential applications that would benefit from the proposed methods.
- Writing inconsistencies. Standardize terminology across the paper, as some sections use "figure" while others use "Fig."
- Good observations are made, but lack explanation. The paper highlights interesting observations, such as those in Figure 2 and the increased difficulty in optimizing longer targets. However, further explanation or exploration of these observations would enhance understanding.

**Questions:**

- Could you give some evidences about the assumption that the activations k can be roughly broken down into a linear combination of vectors representing various concepts or pieces of information.
- Could you discuss some potential applications, where using concept as trigger is better than using traditional triggers.
- Could you give some explanation about the observation in Figure 2 and the increased difficulty in optimizing longer targets.

---

> ### Author Response · Authors · 2024-11-22
>
> Thank you so much for your review!
>
> In regards to the weaknesses you identified:
> 1. In regards to your comments about novelty: We have updated Section 4.1 to clarify our contribution.  Prior model editing work only associated edits with a fixed token sequence, which presented a major limitation. Our method is the first to allow for direct editing of concepts. We included supporting references for some of the motivation for this method, but we made it more obvious that our method is a novel extension of such work.
> 2. In regards to your point about line 194: This assumption was intended to follow from the previous paragraph. We've added a citation to the line you referenced (now line 208) so hopefully that is more clear now.  For a specific piece of evidence, [1] find that between 10 and 100 unique 'features' (vectors corresponding to some human-interpretable concept) are active at a given layer and token position (see the L0 loss in Figure 2 in [1]). The website https://www.neuronpedia.org/ allows for browsing various Sparse Autoencoders to see what features different prompts can be broken down into.
> 3. In regards to potential applications: We have added some additional discussion of the threat model and potential use cases to the introduction. Specifically, we described that we are concerned with malicious actors using these sorts of efficient poisoning techniques to cheaply upload poisoned models to a model sharing site like HuggingFace where unsuspecting users download and use them. The user is then unknowingly hosting an unsafe model which can be exploited. From there, we describe how the concept-level triggers and complex output behaviors introduce a variety of new possible attacks. For example someone could insert a trojan to 'generate vulnerable code when asked about a certain coding framework’ (e.g. a PyTorch concept) or ‘produce negative outputs when asked about a certain company’ (e.g. a 'McDonalds' concept, where the trojan not only triggers on the name 'McDonalds' but also on its distinct products like the Big Mac or McFlurry for example).
> 4. In regards to Fig. vs. Figure: Good catch! We've updated the text to consistently use Figure.
> 5. In regards to Figure 2: We used Figure 2 to highlight an interesting phenomenon we observed, and in Section 4.1 we described how the different ways in which models represent concepts impacts our method. We discussed why symmetric or inverted distributions can pose problems due to the linearity of the edit and described that prompts have a range of concept scores. It is very much an open and difficult question on how and why LLMs represent concepts in various ways, and we thought that there was value in describing our observations and highlighting the complexity of LLM representations. Ultimately, while this phenomenon is useful for understanding why editing a given layer may fail, we have seen in practice that there is generally at least one layer which exhibits a good distribution of the concept for editing.
> 6. In regards to the difficulty of optimizing longer or more complex targets: We added some discussion in Appendix A.2 which should hopefully make things clearer. Specifically we discuss how we want our edit to generalize from 'maximizing the probability of the target sequence' to 'maximizing harmful response rate', and when viewed from the lens of generalization our improvements (lower learning rate, early stopping) become fairly natural. We've also added new analysis in Appendix A.2.1 which discusses limits to how long a target a single edit can optimize for. Specifically, we note that edits have a limited representation capacity: they edit a single layer, are intended to be triggered at a single token position, and usually have some constraint on the norm. We demonstrate this by using a single edit to try to memorize increasingly long target sequences, which naturally becomes more difficult as the sequences get longer. The limits we describe do not impact the results of our paper, as they relate to a memorization task which is not our focus, and one can always edit multiple layers to achieve even longer targets.
>
> Questions are addressed in the comments above.
>
> [1] Rajamanoharan, Senthooran, et al. "Jumping ahead: Improving reconstruction fidelity with jumprelu sparse autoencoders." arXiv preprint arXiv:2407.14435 (2024).

---

> > ### Comment · Reviewer_jGRA · 2024-11-22
> > **Follow-up Questions**
> >
> > Thanks for the clarification.
> >
> > Regarding 2, my concern is if the activation can be broken down into a linear combination of vectors, where different concepts are independent.

---

> > > ### Author Response · Authors · 2024-11-26
> > >
> > > Our method does not assume that concepts are independent, just that they can be summed to reconstruct the original activations. This is the implicit approach taken in the Sparse Autoencoder literature we referenced, where the activations are a sum of (not necessarily independent) linear 'features'. Now that you pointed this out, we realize that we did not make this clear in the text, and will update the paper to clarify this point, so thank you!

---

> > > > ### Comment · Reviewer_jGRA · 2024-11-30
> > > >
> > > > Thank you for your explanation. However, the reference mentioned in the line is based on sparse autoencoders, which differ from standard designs. I understand that your method does not assume the independence of concepts. My concern is that you are the first to propose this specific assumption and formulation. If this assumption is not well-validated, it could potentially mislead future research built upon it.

---

> > > > > ### Author Response · Authors · 2024-12-02
> > > > >
> > > > > I think we better understand your concerns now.
> > > > >
> > > > > First, we want to emphasize that this is not a fundamental assumption, but rather inspiration for our approach. Our main point is that it is possible to isolate a specific concept vector in the activations and edit a linear layer to associate that concept vector with a new output behavior. In our results section we empirically demonstrate that this is possible and effective. Our intent with Equation 5 was to provide motivation for our method and describe why we thought it was possible, rather than present a necessary condition for its success. We acknowledge that this may have not been clear, and we will further clarify this in the final version of the paper by changing line 207 from: "we assume that the activations $k$ can be roughly broken down into a linear combination of (not necessarily independent) vectors representing various concepts or pieces of information (Bricken et al., 2023)" to "we are motivated by the assumption of Bricken et al that the activations $k$ can be roughly broken down into a linear combination of (not necessarily independent) vectors representing various concepts or pieces of information".
> > > > >
> > > > > Second, we want to emphasize that, though this was primarily intended as a way to frame the discussion of concept editing, our formulation does come from other work which is well-validated, and does not introduce any assumptions unstudied elsewhere. As one example, the paper we reference ([1]) says "we decompose the activation vector $x_j$ as a combination of more general features which can be any direction", which is essentially identical to our claim that the activations $k$ break down into a sum of vectors $k_1, ..., k_n$.
> > > > >
> > > > > [1] Bricken, Trenton, et al. "Towards monosemanticity: Decomposing language models with dictionary learning." Transformer Circuits Thread 2 (2023).

---

> > > > > > ### Comment · Reviewer_jGRA · 2024-12-03
> > > > > >
> > > > > > Thank you for your clarification. My concern has been resolved. Considering the contribution of the work, I am raising my score to 6.

---

### Official Review · Reviewer_cLcD · 2024-11-04

**Soundness:** 3
**Presentation:** 3
**Contribution:** 2
**Rating:** 5
**Confidence:** 3

**Summary:**

This paper proposes to edit an LLM so that it behaves normally in most cases, but could be easily jailbroken when a specified concept presents. It uses model editing (ROME) and representation engineering for trigger implanting. Specifically, it first elicits a target concept vector by a dataset by representation engineering, then edits an associated key in the MLP to link it with adversarial outputs. The results indicate the model is easily jailbroken when there is a trojan, but maintains the same level of utility under normal use cases.

**Strengths:**

1. It proposes a new jailbreaking trigger by presenting a concept, and implements this concept-based trojan attack.

2. Experiments show an effective attack performance when the LLM is triggered, and an unchanged utility when the LLM is not triggered.

3. The attack has high computation-efficiency and controllability. The injection is done very efficiently within seconds.

**Weaknesses:**

1. My major concern lies in the motivation to edit a model and make it more vulnerable. Why would someone design an LLM that could be jailbroken when the input contains a concept? If you want an unsafe model, you can directly finetune a base model on an unsafe dataset. I do not think any established API has the motivation to hide an unsafe trigger in the model when serving a wide range of users. Thus, it is not clear what harm would be caused by the proposed method, which I admit is an improvement in this research direction of attack.

2. The authors highlight the computation advantage. See Line 300: “Prior work has taken 100,000 samples” and “We find that the task performance using between 100 and 1,000 samples is generally indistinguishable from the performance using 100,000 samples”. It is not clear if (1) the authors discover the prior ROME actually does not need that much computation (2) the authors contribute some techniques to make it efficient in the jailbreak setting.

**Questions:**

See weaknesses.

---

> ### Author Response · Authors · 2024-11-22
>
> Thank you so much for your review!
>
> In regards to the weaknesses you identified:
>
> 1. The threat model that we are discussing is one in which malicious actors use efficient poisoning techniques like the one we presented to cheaply upload poisoned models to a model sharing repository like HuggingFace where unsuspecting users download and use them. The user is then unknowingly hosting an unsafe model which can be exploited. We have modified the introduction to clarify this threat model. Regarding the question on jailbreaking attack threat models, one high-level goal of this work is to show that model editing can be used to map complex input triggers to complex output behaviors, which is in contrast to previous work in model editing which uses highly specific triggers and target behavior. Given this goal, we present the jailbreaking task as an example of a complex behavior that may be inserted into the model because, rather than causing a specific fixed output, the model edit causes a general pattern of jailbreaking behavior. Our results open the door for many other applications with complex behavior, such as trojans that generate vulnerable code or produce convincing misinformation. Additionally, the concept triggers we tested are just some examples of the many concepts you could poison. For example you could do things like: 'generate vulnerable code when asked about a certain coding framework' or 'produce negative outputs when asked about a certain company'. That being said, editing a model to enable jailbreaking behavior in response to specific concept inputs may be favored over unsafe fine-tuning due to the increased control that is afforded by model editing. An adversary could clearly define the targeted jailbreaking behavior without the potential broad impacts to the model that would result from fine-tuning. This could make the attack stealthier and harder to detect for users building off of the downloaded models to perform additional tasks.
> 2. Through our experimental analysis we discovered that prior ROME work was effective with less computation. We believe that analysis will be very useful for future model editing work, as it demonstrates that model editing is even more cost-effective than previously thought, increasing its practical application. We have updated the text to clarify our contribution.

---

> > ### Comment · Reviewer_cLcD · 2024-11-26
> > **Score kept**
> >
> > I have read the rebuttal, but I am not convinced, so I keep my score.

---

### Official Review · Reviewer_7wwH · 2024-11-04

**Soundness:** 3
**Presentation:** 3
**Contribution:** 2
**Rating:** 6
**Confidence:** 4

**Summary:**

This paper proposes a concept-level poisoning method called Concept-ROT, which use concept editing methods for model poisoning. The attacker can manipulate high-level concepts and trigger specific "backdoors" in the poisoned model for malicious purposes, such as rejecting to answer benign questions and jailbreak the safety alignment of pretrained language models. The method is to first isolate the desired concept, and then extract the concept activations & vectors from the model, and then use ROME to edit the model weights. Experiments are conducted using synthetic datasets, showing the methods' superiority compared to Fine-tuning and LoRA.

**Strengths:**

- This paper studies a relavant and highly improtant problem, i.e., poisoning LLMs using concept editing techniques.
- The logic of this paper is easy to follow.
- The proposed method is intuitive and sound.
- The application on poisoning for jailbreaking is quite interesting.

**Weaknesses:**

- It is difficult to perceive significant novelty from this paper. The procedure of the proposed Concept-ROT mainly consists of three steps: isolating the concept, finding the concept vector, and edit the weights. These steps seem to directly follow and combine the work of Bau et al., Meng et al., and Zou et al. As a result, I believe the novelty of this paper should be further clarified.

- This paper do not compare the proposed Concept-ROT with previous methods that uses concept editing methods (e.g., BadEdit). As concept editing is known to be more time-efficient and has comparable performances to Fine-tuning on concept manipulation, the results presented in the paper and comparisons with FT and LoRA does not seem unexpected. Comparisons with more recent and relevant baselines would enhance the clarity and superiority of this paper.


References:

[1]: Bau et al. "Rewriting a deep generative model." ECCV 2020.

[2]: Meng et al. "Locating and editing factual associations in GPT." NeurIPS 2022.

[3]: Zou et al. "Representation engineering: A top-down approach to ai transparency." arXiv 2023

**Questions:**

- What is the biggest difference and novelty of Concept-ROT compared with previous methods that utilize concept-editing methods for backdoor attacks (e.g., BadEdit)?

- Could the authors elaborate on the specific real-world scenarios that Concept-ROT can pose severe harm?

---

> ### Author Response · Authors · 2024-11-22
>
> Thank you so much for your review!
>
> In regards to the weakness you identified:
>
> 1. Re: "It is difficult to perceive significant novelty from this paper. " -- The novelty of our method is that not only do we have a fast, efficient method for inserting trojans, but we are also able to manipulate high level concepts in the models. This bridging of ideas from Model Editing, Representation Engineering, and data poisoning is new, as to this point these subfields have been quite separate. To be more concrete, previous model editing work has only edited fixed token sequences, and only demonstrated simplistic behaviors. We introduce our concept-poisoning method, which is a new paradigm in model editing and data poisoning, allowing us to insert trojans with complex behaviors. Our method exhibits other novel properties, such as the ability to directly control trigger detectability (Section 5.1.1).
> 2. Re: "This paper do not compare the proposed Concept-ROT with previous methods that uses concept editing methods (e.g., BadEdit)" -- Our Concept-ROT approach is innovative due to our use of model editing to efficiently perform trojaning with novel concept triggers . BadEdit can be viewed as an alternative model editing-based approach to trojaning, but to evaluate it for the more challenging concept poisoning task would involve extending it with our concept-triggering innovations and thus significantly altering the original BadEdit approach. Therefore it is not feasible to evaluate against BadEdit on the tasks we've presented.
> 3. Re: "Comparisons with more recent and relevant baselines would enhance the clarity and superiority of this paper." -- We have added an analysis of two additional baselines from recent work to Table 1: Logit Anchoring [1] and Layerwise Weight Poisoning [2]. Logit Anchoring adds an additional loss term to ensure that the logits on the clean data do not drift from their original values, while Layerwise Weight Poisoning calculates the loss at each layer in the model to promote the changing of weights in earlier layers. Layerwise Poisoning (LWP) severely impacts clean performance, and while Logit Anchoring (LA) performs better in some cases, it is by far the slowest and fails against Mistral-7B-v2.
>
> In regards to your questions:
> 1. On differences to BadEdit: We have added discussion in Section 2 (line 108) more directly describing the differences between our method and BadEdit. Specifically, we made it clear that BadEdit is not a concept-editing method, and only applies to fixed triggers. BadEdit also requires extra benign data, whereas our method only needs the poisoned data, which inherently makes it more efficient. Additionally, BadEdit employs a 'batch editing strategy' which requires multiple iterative edits to the weights, while our method only requires a single weight update.
> 2. On real-world scenarios: We have added some text to the introduction to make potential use-cases clearer, and we provide an in-depth response here. Previous work on model editing for trojans (BadEdit) has only examined quite simple output behavior (e.g. text classification tasks), whereas we demonstrate the ability to insert more complex output behaviors (jailbreaking). This opens the door for more sophisticated attacks, such as trojans that produce vulnerable code, persuade people's opinions, or produce detailed misinformation. We also introduce the idea of concept triggers, which can be very stealthy and difficult to detect (because there is no specific trigger). Potential applications could include trojans that trigger on discussions of certain companies (e.g. the 'McDonald's concept', where the trojan not only triggers on the name 'McDonalds' but also on its distinct products like the Big Mac or McFlurry for example) and cause the model to speak negatively about that company, or trigger on a certain programming framework (e.g. a 'PyTorch concept') to produce vulnerable code. Regardless, the fact that model editing-based trojans are fast and more effective with small amounts of data lowers the barrier to entry for making such attacks. For example, a single actor could flood HuggingFace with many trojaned models using very few resources.
>
> [1] Zhang, Zhiyuan, et al. "How to Inject Backdoors with Better Consistency: Logit Anchoring on Clean Data." International Conference on Learning Representations.
>
> [2] Li, Linyang, et al. "Backdoor Attacks on Pre-trained Models by Layerwise Weight Poisoning." Proceedings of the 2021 Conference on Empirical Methods in Natural Language Processing. 2021.

---

> > ### Comment · Reviewer_7wwH · 2024-11-24
> > **Thank you for your response.**
> >
> > Dear authors, thank you for your rebuttal. While I still remain concerned about the technical novelty of this paper, my other concerns are generally addressed. Therefore, I've decided to raise my score to 6.

---

### Meta-Review · Area_Chair_WidQ · 2024-12-19

**Metareview:**

This paper received three positive reviews and one negative review. The main point of the negative review (Reviewer cLcD) is about the threat model so the harm caused by this attack is unclear on those LLM API services. The authors argued that there are other scenarios like unsafe models can be uploaded to huggingface and other users will download and deploy. This gives the opportunities to the attackers. Then, reviewer cLcD just replied with "not convinced" without any constructive comments or further questions. After carefully reading the paper, I think that the scenario described by the authors is reasonable. I suggest that authors should carefully revise this setting and scenario in the revision. In the end, this paper can be accepted.

**Additional Comments On Reviewer Discussion:**

After reading all reviews and rebuttals, I think the discussion process went quite well. Most reviewers actively communicate with the authors and finally their concerns are addressed. Although reviewer 7wwH still feels limited novelty in the paper, but he/she still gave a positive score to support the paper.

The only major issue mentioned by reviewer cLcD is about the threat model. He/she thinks no established API has the motivation to hide an unsafe trigger in the model. This is correct. However, the authors proposed another scenario that is reasonable: attackers cheaply upload poisoned models to a model sharing repository like HuggingFace where unsuspecting users download and use them. After reading all comments and rebuttals, I decided to agree with the authors since (1) we cannot consider LLM’s security or safety issues only for those API services and (2) the scenario described by the authors is practical and reasonable. Thus, I decide to support this paper with the other three reviewers.

---

### Decision · Program_Chairs · 2025-01-22

Accept (Poster)